# Predicting the Current and Future Distributions of *Frankliniella occidentalis* (Pergande) Based on the MaxEnt Species Distribution Model

**DOI:** 10.3390/insects14050458

**Published:** 2023-05-12

**Authors:** Zhiling Wang, Danping Xu, Wenkai Liao, Yan Xu, Zhihang Zhuo

**Affiliations:** College of Life Science, China West Normal University, Nanchong 637002, China

**Keywords:** invasive species, MaxEnt model, *Frankliniella occidentalis* (Pergande)

## Abstract

**Simple Summary:**

*Frankliniella occidentalis* (Pergande) is an invasive pest of a small size with a high reproduction rate. The main damage caused by *F. occidentalis* is the transmission of the tomato spotted wilt virus, which has had certain impacts on agriculture in China. In this work, the maximum entropy model (MaxEnt) in the species distribution model was used to predict the suitable existence range of *F. occidentalis* in China, and areas of three levels, including high-, medium- and low-suitable zones, were obtained. In particular, the trend of the species’ future geographical distribution was derived based on the results of future projections. The results show that the areas of high- and low-suitability zones will continue to decrease in the future, and the area of medium-suitability zones will significantly increase. Then, the environmental variables affecting the distribution of the insect were screened, and it was concluded that temperature and precipitation might be the two factors affecting the distribution of this species that are worthy of consideration. This study provides certain references for the future pest control of *F. occidentalis*.

**Abstract:**

Climate change has a highly significant impact on the distribution of species. As the greenhouse effect intensifies each year, the distribution of organisms responds to this challenge in diverse ways. Therefore, climatic environmental variables are a key entry point for capturing the current and future distribution trends of pests. *Frankliniella occidentalis* is an invasive pest attested worldwide. Its damage is mainly divided into two aspects, including mechanical damage caused by its feeding and egg laying and the spread of tomato spotted wilt virus (TSWV). TSWV is the most dominant transmitted virulent disease. Moreover, *F. occidentalis* is the major vector for the transmission of this virus, which poses a grave threat to the yield and survival of our crops. In this study, the distribution pattern of this pest was explored using 19 bioclimatic variables based on the Maxent model. The results indicated that in the future, high-suitability areas of *F. occidentalis* will be widely distributed in 19 provinces of China, with Hebei, Henan, Shandong, Tianjin and Yunnan being the most abundant. Among the 19 bioclimatic variables, the five variables of annual mean temperature (Bio 1), temperature seasonality (standard deviation × 100) (Bio 4), min temperature of the coldest month (Bio 6), mean temperature of the driest quarter (Bio 9) and precipitation of the coldest quarter (Bio 19) were selected as the key environmental variables affecting the distribution of *F. occidentalis*. In summary, temperature and precipitation are vital factors for the study of the species’ distribution, and this study aims to provide new perspectives for the control of this pest in China.

## 1. Introduction

*Frankliniella occidentalis* belongs to the genus Thrips in the family Thripsidae of the suborder Terebrantia. It is a greenhouse pest attested worldwide and an especially critical quarantine target in various countries [1,2]. It spread rapidly around the world from the 1970s to 1980s, causing serious economic damage in many countries. *Frankliniella occidentalis* was first discovered in California in 1895, and since the 1970s, it has invaded most of the world and become the dominant population in the majority of areas where it has been introduced [3]. In China, *F. occidentalis* was first intercepted in 2000 in Kunming, Yunnan Province, and the establishment of a population was first reported in 2003 in Beijing [4]. The host plants of *F. occidentalis* cover more than 60 families and 500 species and are especially hazardous for economic crops such as vegetables, flowers, fruit trees and cotton. Furthermore, the species’ biological and ecological characteristics, such as its feeding and hidden nature of feeding, residence, ability to spread easily, relatively short generation development period, and strong reproductive capacity, as well as the R-strategy adopted by the population, make it more difficult to control [5,6]. *Frankliniella occidentalis* causes damage to leaves, flowers, and fruits by puncturing and grasping the flower and leaf tissues with its mouthparts to suck the sap and then extracting the contents of the ruptured cells during the feeding process [7,8,9]. However, the more serious damage caused by this insect is reflected in the transmission of TSWV, as compared to the mechanical damage caused to plants. TSWV is a tripartite negative/ambisense ssRNA virus of the genus Orthotospovirus, the family Tospoviridae and the order Bunyavirales [10,11,12,13]. It is noteworthy that TSWV is transmitted only through the species of a single family, Thripidae, of which *F. occidentalis*, studied here, is the main and most efficient vector [14].

Pests and diseases are the main problems faced during crop cultivation, and the prediction of the potential geographical distribution of pests and diseases is one of the major areas of study for the quantitative risk assessment of pests [15]. Species distribution models (SDM) are an invaluable tool for studying species suitability, predicting the probability that a target species will be present at a given location, or quantifying habitat suitability as a function of multiple predictor variables [16,17]. The software commonly used for prediction include CLIMEX, BIOCLIM, GARP, and MAXENT. In the mid-80s of the 20th century, with the gradual development of SDM technology, a quantitative basis for the analysis of SDM in animals and plants was established for the first time. The Maxent model has shown significant superiority over other models in subsequent practice. The Maxent model is able to analyze the effects of different environmental variables on the suitability distribution of the studied species through a unique response curve. The Maxent model yields excellent statistics, benefiting from the spatial autocorrelation of data, allowing its complex response curves to be fitted and for the model to be remarkably close to the calibration data [18]. Maxent is a standalone Java application for SDM, which is a spatial distribution model of species on geographic scales based on the theory of maximum entropy. In principle, the theory of maximum entropy finds a marginal fitness function for each variable matching the empirical data that is the most informative and whose mean is equal to the mean of the empirical data. Nevertheless, this process can be affected the problem of over-fitting of the model input data. To address this problem, Maxent constrains the modeled distribution to lie within a certain interval around the empirical mean, rather than matching the empirical mean exactly, through a process of regularization [19]. In addition, Maxent can predict the future distribution of species from a few species occurrence records combined with simulated environmental variables. Moreover, it can effectively handle sparse and irregular sampling data and small positioning errors, and its simulation accuracy is far greater than that of other models [20,21,22]. Maxent, in terms of its predictive performance, consistently competes with the top-performing methods. Its ability to predict the range of species distribution using two locations of known species occurrence and non-occurrence has been validated with the presence–absence model. Since Maxent’s inception, the model has been widely used in ecology, evolution, biosecurity, etc., to explore the distribution of species, map current distribution and predict potential future distributions [18,23].

To date, there have been many studies on the invasive pest *F. occidentalis*, but none of them have involved the prediction of its potential distribution area [4,24]. In this work, the geographical distribution of the species was determined based on the MaxEnt model and linked to 19 bioclimatic variables to analyze the relevance of each variable. Several variables with decisive effects were identified to predict the future distribution of the species. This is empirical, data-driven research, and its aim is to provide an important reference and theoretical basis for the development of reasonable prevention and control measures.

## 2. Materials and Methods

### 2.1. Data

#### 2.1.1. Species Distribution Data

According to the needs of Maxent modeling, the data are divided into two parts, namely, species occurrence data and environmental data, and the species occurrence data comprise the latitude and longitude information of the “presence” points. Species occurrence data for *F. occidentalis* were obtained from the Global Biodiversity Information Facility (http://www.gbif.org, accessed on 12 January 2023) and the literature. When exact latitude and longitude geographic coordinates were not available in the occurrence records, imprecise locations were excluded from the analysis. Google Earth (http://ditu.google.cn/, accessed on 12 January 2023) was used to determine the longitude and latitude of the occurrence sites for which only locations were provided in the literature. The ENMTools version 1.0.4 of the R platform was used to trim the distribution point according to the longitude; thus, the occurrence points obtained were necessarily smaller than the actual distribution area. Nonetheless, the Maxent model selected in this work still has the characteristics of excellent prediction and accuracy in the case of a small sample size.

#### 2.1.2. Bioclimatic Variables

The 19 bioclimatic variables used in this study were downloaded from the WorldClim-Global climate database (http//www.worldclim.org/download, accessed on 12 January 2023), and averaged data were obtained for each environmental variable from 1970 to 2000, including present and future climatic conditions. The global climate model data were based on CMIP5 (IPCC Fifth Assessment Report, AR5) and the bioclimatic data were downloaded from WorldClim version 2.1 to represent the current climate. To estimate the impact of possible future climatic conditions on the distribution of *F. occidentalis*, the atmospheric circulation model CCCma_CanESM2 was used in CMIP5 to simulate the RCPs. In this study, three of the RCPs, including RCP2.6, RCP4.5, and RCP8.5, were selected for future projections of the 2050s and 2090s under three different scenario models. RCP2.6 is a very mild scenario representing a future with low emissions and a focus on sustainable development. RCP4.5 is a moderate-emission scenario representing a relatively mild future. RCP8.5 is a high-emission scenario representing a future of high dependence on fossil fuels. The spatial resolution of the variables was 2.5 arc-minutes (approximately 4.5 km^2^).

### 2.2. Data Analysis

#### Bioclimatic Variable Selection

The Maxent model provides two metrics to determine the importance of the output climate variable in the final model, namely, the percentage of contribution and ranking importance. Each step of the Maxent algorithm increases the gain of the model by modifying the individual variable covariates, and the program can then allocate the increase to the climate variables on which the variables depend. The percentage contribution was obtained at the end of the training process by means of transforming the increase into a percentage. The permutation importance of each variable was determined by randomly arranging the value of that variable between training points and measuring the reduction in the training results of the receiver operating characteristic curve (ROC). A large reduction indicates that the model relies heavily on the given variable [21]. Based on the model-generated fitness index, the suitable distribution area of *F. occidentalis* was divided into four levels, namely, the unsuitable area, moderately suitable area, poorly suitable area and highly suitable area. A 10-fold repeat cross-validation was then performed to run the Maxent software in order to prevent random errors resulting from the predicted sample. Finally, the reclassification function in ArcGIS software was used to obtain the distribution level of *F. occidentalis* in the national suitable area, as shown in Figure 1.

Due to the correlation between environmental factors and species distribution, not all variables have an impact on pest distribution; thus, it is necessary to screen environmental variables. Maxent software was used to perform the modeling and variable selection. The jack-knife test was employed to obtain the contribution of the variables, and Pearson’s correlation coefficient in SPSS 22 software was adopted to calculate the correlation between variables to minimize multicollinearity in the analysis. Variables with coefficients *r* ≥ |0.8| were removed from the initial environment dataset. Moreover, the variables that contributed less to the prediction results of Maxent model were eliminated to improve the simulation accuracy of the model. The Maxent model of *F. occidentalis*’ distribution was reconstructed on this basis, and the accuracy of the simulated results was evaluated.

### 2.3. Evaluation Criteria

In the Maxent model, the AUC (area under the curve) of the ROC curve is a commonly used threshold-independent measure that is used as the default evaluation criterion for evaluating the accuracy of the model [23]. The Maxent software can automatically generate the ROC curve and provide the corresponding AUC value. AUC is a threshold-independent assessment criterion that is not sensitive to species prevalence and is widely used for niche modeling. Generally, the range of the AUC value is 0–1, and the higher the value is, the better the prediction effect of the model is. According to the AUC value, the prediction accuracy of the model can be divided into five levels: 0–0.6, failure; 0.6–0.7, poor; 0.7–0.8, general; 0.8–0.9, good; and 0.9–1, excellent [25].

## 3. Results

### 3.1. Key Environment Variable Selection and Model Performance

The contribution percentage is an extremely important metric used by Maxent to determine the input variables in the final model. Variational contributions to each of the environmental variables can be obtained by plugging the species occurrence point data into the Maxent software modeling and passing the jack-knife test. The results obtained in this work show that the annual mean temperature (Bio 1) contributed the most to the distribution, (47.9%), followed by temperature seasonality (standard deviation × 100) (Bio 4) (22.8%) and the min temperature of the coldest month (Bio 6) (21.9%). The various values of the mean temperature of the driest quarter (Bio 9) (6.6%) and precipitation of the coldest quarter (Bio 19) (0.9%) only accounted for a minor portion. The cumulative value of the top three reached 92.6%, which means that Bio 1, Bio 4 and Bio 6 are the main environmental variables affecting the habitat suitability of *F. occidentalis*, which can well reflect the potential habitat of this species. If there is a high correlation between variables, this may interfere with the model’s performance; thus, highly correlated variables are filtered out. This study used 19 climate variables combined with the Maxent model to predict the potential distribution of *F. occidentalis* across the country. As shown in Table 1, based on the results, we finally screened out five important environmental variables, namely Bio 1, Bio 4, Bio 6, Bio 9 and Bio 19, and the highest collinearity was between variables Bio 4 and Bio 6, with a positive correlation of 75.5%. It is worth noting that the correlation coefficient between all the variables was below 0.8, which indicates that there is no collinearity or low collinearity between them.

In this paper, the complexity of the model is controlled by adding parameters to the ENMTools function. The complexity of the Maxent model depends on two parameters: the model features and the regularization of the parameter settings. The Maxent model is based on a set of “features”, such as the linear (L), hinge (H), quadratic (Q), threshold (T), and product (P) features. Given site-specific environmental values, the model is able to determine the predicted probability of existence or the rate of environmental suitability of the study area. The feature combination (FC) settings for this study start with the default “LQ” training model, which is optimized to the “LQH” training model. The regularization multiplier (RM) value is optimized from the default value of 1 to 3.0, and the AUC value is optimized from 0.742, under the default setting, to 0.888. Figure 2 is the ROC curve of the same data used to estimate the model prediction accuracy, and the number of replicates of the first run of the model is 10 times. The average test AUC of repeated runs is 0.888, and the standard deviation is 0.044. These results show that the model runs well, and the prediction accuracy is high.

### 3.2. Species Response and Potential Habitat Suitability Distribution

The responses of the species to changes in the selected variables are shown in Figure 3, and each curve represents a different model, i.e., a Maxent model created using only the corresponding variable. These plots reflect the dependence of the prediction suitability on the selected variable and the dependence caused by the correlation between the selected variable and other variables. According to the response curve, when the annual mean temperature (Bio 1) is between 12.0 °C and 13.0 °C, this is most suitable for the survival of *F. occidentali*, and the curve shows a straight downward trend between 13.0 °C and 24.0 °C and remains stable after 24.0 °C. It is noteworthy that the min temperature of the coldest month (Bio 6) has two peaks of 0.80 and 0.88 at −11.0 °C and 5.4 °C, respectively, as can be seen from Figure 3b. In between, at −1.78 °C, there is a small, low peak with a value of 0.69. It can be seen from Figure 3c that when the mean temperature of the driest quarter (Bio 9) is not between 1.9 °C and 5.2 °C, there is a stagnation stage, and then the temperature gradually increases by 9.8 °C to reach a peak. Overall, increased temperatures lead to a greater distribution. In addition, the trend of the precipitation of the coldest quarter (Bio 19) variable has a rapid rise and a rapid decline, peaking at 18.3 mm of precipitation and turning from a rapid decline to flat decline when the precipitation is 76.17 mm. It can be seen that temperature and precipitation are the main factors affecting the distribution of *F. occidentalis*.

As shown in Figure 1, *F. occidentalis* is distributed in China’s plateau temperate, middle-temperate, warm-temperate, plateau subtropical and southeast subtropical zones. The high-suitable zone is mainly distributed in Hebei, Henan, Shandong, Tianjin and Yunnan Provinces. Table 2 shows that the total area of *F. occidentalis* is currently 471,630 km^2^, accounting for 4.8% of China’s area. Hebei has the largest area of 92,700 km^2^, accounting for 14.0% of the total area of high-suitability areas in China. Henan, Shandong, Tianjin and Yunnan account for 9.7%, 12.6%, 1.8% and 3.2% of the total area of high suitability, respectively.

### 3.3. Environmental Adaptation under Current and Future Climate Scenarios

In this study, the potential distribution of *F. occidentalis* in the context of two future periods (2050s and 2090s) and three RCPs (RCP2.6, RCP4.5 and RCP8.5) was plotted, and the potential distribution level is shown in Figure 4. Table 3 shows that future climate change will have a certain impact on the potential distribution area of *F. occidentalis*, and the level of ecologically suitable zones will change greatly. In the 2050s RCP2.6, RCP4.5 and RCP8.5 scenarios, the area of the high-suitability zone for *F. occidentalis* is 66.15 × 10^4^ km^2^, 60.72 × 10^4^ km^2^ and 68.65 × 10^4^ km^2^, respectively; in the 2090s RCP2.6, RCP4.5 and RCP8.5 scenarios, the area of the high-suitability zone for *F. occidentalis* is 67.32 × 10^4^ km^2^ and 66.37 × 10^4^ km^2^, respectively. In the RCP4.5 and RCP8.5 scenarios, the area of the *F. occidentalis* high-suitability zone is 67.32 × 10^4^ km^2^, 66.37 × 10^4^ km^2^ and 59.40 × 10^4^ km^2^, respectively. It can be seen that in the different RCP scenarios, the area of the high-suitability area shows an overall decreasing trend. In the RCP8.5 scenario in the 2050s, the area of the medium-suitable area is increased the most, with an increase of 23.4% compared with the current scenario. It is expected that in the RCP2.6, RCP4.5 and RCP8.5 scenarios in the 2090s, the total suitable area will decrease by 6.3%, 18.4% and 6.6%, respectively. In general, the area of the high- and low-suitable areas will continue to decrease in the future, and the area of the medium-suitable areas will significantly increase.

## 4. Discussion

The Maxent model was first employed to predict the potential distribution of *F. occidentalis* in China for the current and future periods in this work. The potential distribution of *F. occidentalis* was modeled by simulating two future periods (2050s and 2090s) and three RCP scenarios (RCP2.6, RCP4.5 and RCP8.5) scenarios. Under different sampling assumptions, Maxent is better suited for predictive work using models with relative abundance or in helping to identify areas with large, stable populations, meaning that further consideration is needed in the selection of predicted species [20,26]. Nevertheless, the selection of the target species for this paper took this issue into account, and due to the relatively rich distribution of *F. occidentalis*, the impact on the prediction of this species is minimal. Maxent is limited to the analysis of abiotic factors, such as temperature and precipitation, and the results output by the model do not consider the influences of biological factors on species distribution. The combination of abiotic and biotic factors will be considered in future prediction work, and this is an area that deserves attention and reflection in future forecasting efforts [27].

*Frankliniella occidentalis* is an invader of agricultural and horticultural crops. The damage caused by the pest is primarily in the form of direct damage caused by feeding and egg laying, as well as extensive crop damage through the spread of destructive plant viruses [24]. The control of the rate of development, reproductive capacity and other growth distribution patterns of this destructive, invasive pest is particularly relevant. It was shown that fluctuations in temperature had significant effects on all these aspects, which, in turn, directly affected the distribution of *F. occidentalis* [24,28,29,30,31]. In this study, the main environmental variables influencing changes in the distribution of *F. occidentalis* were identified based on the results of the jack-knife test and the contribution rate of the climate variables calculated using the Maxent model. We effectively excluded environmental variables with a low contribution rate and high correlation. Five key environmental factors were selected to reconstruct the model, which improved the accuracy of the prediction results. According to the results of Maxent modeling, the future high-suitability zones of *F. occidentalis* will mainly be distributed in Hebei, Henan, Shandong, Tianjin and Yunnan Provinces. This corresponds to the current area of frequent insect infestations. The key environmental variables affecting the distribution, namely Bio 1, Bio 4, Bio 6, Bio 9 and Bio 19, were screened based on their rates of contribution to the probability of occurrence. This suggests that climatic conditions may be an important factor limiting the pest’s distribution, with air temperature (Bio 1, Bio 4, Bio 6) having the greatest influence on the distribution. With the increase in the RCPs, *F. occidentalis* showed a trend of transition from the low-suitability zone and the high-suitability zone to the medium-suitability zone in the simulated future scenarios. It is located in the mid-latitude zone of the subtropical monsoon climate, an area of significant increase. Nationally, the high-suitability zone of *F. occidentalis* is mainly concentrated near the 800 mm annual precipitation line, which is the dividing line between the subtropical monsoon climate and temperate monsoon climate. This is consistent with the previous view that temperature affects the species’ distribution [32,33,34].

The expansion and spread of *F. occidentalis* are mainly due to the indiscriminate use of synthetic pesticides during the last 30 years [11,35]. However, it should be noted that all the distribution points of *F. occidentalis* were downloaded from the Global Biodiversity Information Facility (http://www.gbif.org, accessed on 12 January 2023). The distribution points in the raster were screened during the modeling process. Thus, there are no distribution points primarily based on places that have not been sprayed with pesticides, nor are there large areas of suitable habitats that might be excluded from the model. Therefore, the control of *F. occidentalis* cannot be limited to chemical control alone. According to the high fecundity, short cycle, wide host variety and other characteristics of *F. occidentalis*, we make the following suggestions for its control: (1) focus on prevention, establish an early detection system in areas where harm has not yet occurred, especially cold areas, and identify and interrupt its transmission in a timely manner; and (2) combine treatment and closely observe areas where harm has occurred. Biological control should be carried out as much as possible, mainly through large-scale biological and microbial control. Biological methods such as plant extracts, biocontrol bacteria, biological agents and predatory natural enemies could also be used. It is important to prioritize biological control methods for *F. occidentalis*. In terms of natural enemies, there are many predators and fungi that can effectively control this pest. For example, the predator *Chrysopa pallens* (Rambur) can be used to control *F. occidentalis*, with one *C. pallens* (Rambur) capable of consuming 34 to 41 *F. occidentalis* within 24 h [36]. Specifically, the future range of natural enemies can be predicted and then compared to the future range of *F. occidentalis*. Biological control can be carried out by expanding the suitable distribution area of natural enemies where both are present. As for fungal biological control, this mainly involves the active invasion and infection of the host body, including processes such as host recognition, attachment spore differentiation, penetration of the body wall, immune antagonism and colonization within the host. The host’s death is caused by multiple factors, with the ultimate goal of controlling pests through biological means [36,37,38,39]. 

## 5. Conclusions

The potential geographic distribution of *F. occidentalis* in three climate change scenarios (RCP2.6, RCP4.5 and RCP8.5) in the current and two future periods (2050s and 2090s) were successfully simulated based on the Maxent model. The high-suitability zones are mainly distributed in Hebei, Henan, Shandong, Tianjin and Yunnan Provinces. The main environmental variables affecting the distribution of *F. occidentalis* are the annual mean temperature (Bio 1), temperature seasonal variation coefficient (Bio 4), coldest month minimum temperature (Bio 6), driest season average temperature (Bio 9) and coldest season precipitation (Bio 19). The optimal model includes five climate variables, of which the temperature factor has a greater influence than the precipitation factor. In recent years, there have been frequent occurrences of high temperatures during the summer in China, which are expected to have some impact on this insect. However, due to the biological characteristics of *F. occidentalis* and its R-strategy population, the impact of high-temperature stress on the distribution of *F. occidentalis* in the next few years may not be significant. However, in the long term, temperature increases will lead to the expansion of *F. occidentalis*’ distribution area; the predicted potential suitable distribution area of *F. occidentalis* is in good agreement with its actual occurrence. This study is expected to provide a reference for the prediction, forecasting, and effective prevention and control of *F. occidentalis*.

## Figures and Tables

**Figure 1 insects-14-00458-f001:**
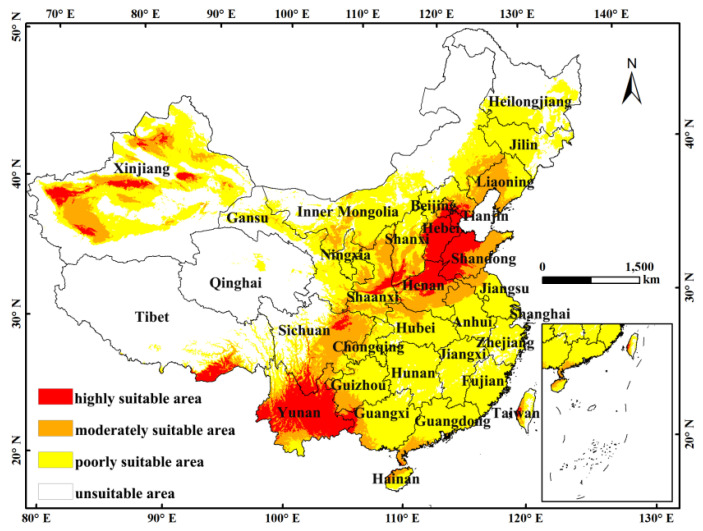
Current suitable climatic distribution of *F. occidentalis* in China. The probability of *F. occidentalis* is shown by the color scale in the area. Red indicates a highly suitable area with a probability higher than 0.66, orange indicates a moderately suitable area with a probability of 0.33–0.66, yellow indicates a poorly suitable area with a probability ranging from 0.05 to 0.33, and white represents unsuitable areas.

**Figure 2 insects-14-00458-f002:**
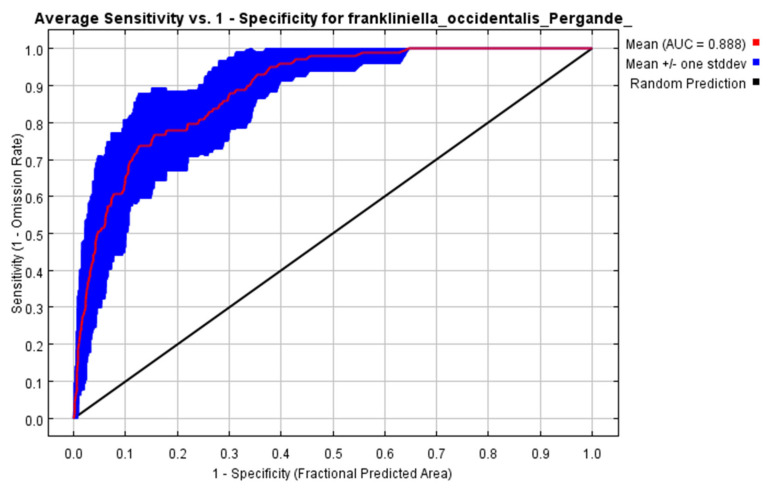
Receiver operating characteristic curve and AUC result of Maxent modeling.

**Figure 3 insects-14-00458-f003:**
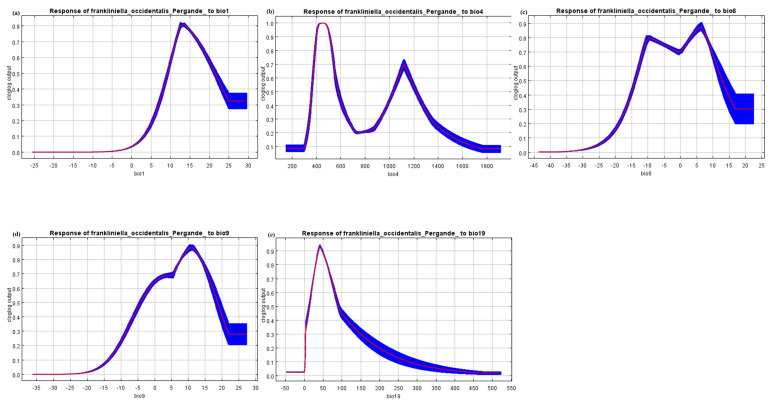
Response curves of the environmental variables that contributed the most to the Maxent models. (**a**) Annual mean temperature (Bio 1). (**b**) Temperature seasonality (standard deviation × 100) (Bio 4). (**c**) Coldest month minimum temperature (Bio 6). (**d**) Driest season average temperature (Bio 9). (**e**) Coldest season precipitation (Bio 19). The red line is the average response of the Maxent run. The blue part is the average +/- one standard deviation.

**Figure 4 insects-14-00458-f004:**
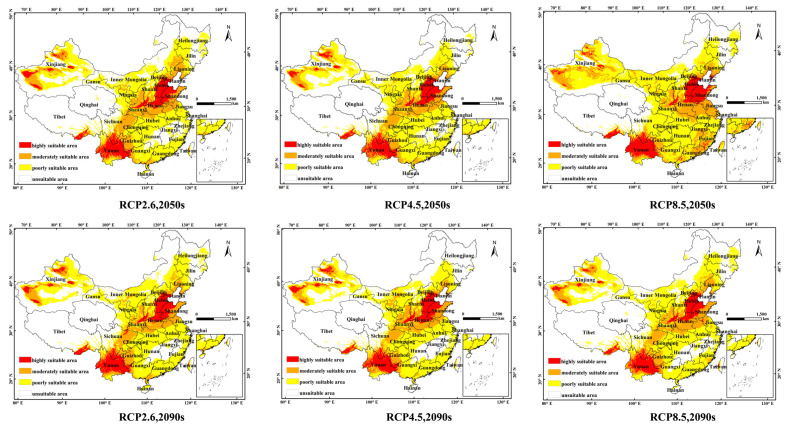
Potential distribution of suitable areas for *F. occidentalis* based on different climate change scenarios in China. The probability of *F. occidentalis* is shown by the color scale in the area. Red indicates a highly suitable area with a probability higher than 0.66, orange indicates a moderately suitable area with a probability of 0.33–0.66, yellow indicates a poorly suitable area with a probability ranging from 0.05 to 0.33, and white represents unsuitable areas.

**Table 1 insects-14-00458-t001:** Pearson correlation coefficients of key environmental factors.

Code	Bio 1	Bio 4	Bio 6	Bio 9
Bio 4	0.704 **			
Bio 6	−0.076	−0.755 **		
Bio 9	0.575 **	−0.139 *	0.712 **	
Bio 19	0.729 **	0.356 **	0.201 **	0.605 **

**, significant correlation at the 0.01 level (bilateral). *, significant correlation at the 0.05 level (bilateral).

**Table 2 insects-14-00458-t002:** Analysis of the main suitable distributions of *F. occidentalis* in China.

Province	High-Suitable Area (10^4^ km^2^)	Total (10^4^ km^2^) *	Percentage of High-Suitable Area in the Province (%)	Percentage of High-Suitable Area in China (%)
Tianjin	1.18	1.20	98.3	1.8
Shandong	8.35	15.58	53.6	12.6
Beijing	0.84	1.64	51.2	1.3
Hebei	9.27	18.88	49.1	14.0
Henan	6.40	16.70	38.3	9.7
Shanxi	1.40	15.67	8.9	2.1
Shaanxi	1.61	20.56	7.8	2.4
Sichuan	3.65	48.60	7.5	5.5
Taiwan	0.20	3.60	5.6	0.3
Yunnan	2.13	39.41	5.4	3.2
Guangxi	1.20	23.76	5.1	1.8
Xinjiang	7.16	166.49	4.3	10.8
Hainan	0.09	3.54	2.5	0.1
Tibet	3.05	122.84	2.5	4.6
Guizhou	0.36	17.62	2.1	0.5
Jiangsu	0.19	10.72	1.8	0.3
Guangdong	0.06	17.97	0.3	0.1
Anhui	0.02	14.01	0.1	0.0
Gansu	0.003	42.58	0.0	0.0
China	47.16	/	/	0.1

* indicates the total area of the corresponding province.

**Table 3 insects-14-00458-t003:** Prediction of the suitable areas for *F. occidentalis* under current and future climatic conditions.

Decade	Scenarios	Predicted Area (10^4^ km^2^)			Comparison with Current Distribution (%)		
		High-Suitable Area	Moderate-Suitable Area	Low-Suitable Area	High-Suitable Area	Moderate-Suitable Area	Low-Suitable Area
Current		66.28	121.93	346.63			
2050s	RCP2.6	66.15	132.87	308.74	−0.2	9.0	−10.9
	RCP4.5	60.72	118.14	308.83	8.4	−3.1	−10.9
	RCP8.5	68.65	150.51	360.55	3.6	23.4	4.0
2090s	RCP2.6	67.32	118.19	324.75	1.6	−3.1	−6.3
	RCP4.5	66.37	115.99	282.98	0.1	−4.9	−18.4
	RCP8.5	59.40	124.11	323.72	−10.4	1.8	−6.6

## Data Availability

The data supporting the results are available in a public repository at: GBIF.org (12 January 2023) GBIF Occurrence Download https://doi.org/10.15468/dl.a6pr65, accessed on 12 January 2023 and Zhiling Wang (2023): *Frankliniella occidentalis* (Pergande) occurrence.xlsx. figshare. Dataset. https://doi.org/10.6084/m9.figshare.21875280.v3, accessed on 18 March 2023.

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
