# Peer review of "Predicting the Current and Future Distributions of Frankliniella occidentalis (Pergande) Based on the MaxEnt Species Distribution Model"

_insects, 2023, doi:10.3390/insects14050458_

Round 1

Reviewer 1 Report

General

Need to check writing by native speaker, some errors ae highlighted in yellow in the manuscript

Several sections in the manuscript digress from its purposes (annotated in the manuscript).

Scientific name of pest should be written in full the first time, later can be abreviated drop author name.

Abstract

short abstract need to provide more specific information regarding results.

Introduction

introduction refers too much to the thrips as a pest but no review is presented related to its situation in China or factor affecting its ecology.

Also, introduction describe much of maxent as a tool.

Should include papers refering to the distribution of this pest () and ecological factors affecting distribution ().

Materials and Methods

Does not mention which specific model is used to make future projections.

Results & Discussion

Streamline tables, pass info to text for tables 1, 2. Main pattern in table 3 is that correlation between variables is less than 0.8, thus transfer main findings to text. 

Include limitations of the work related to: A) just one model was used for future projections, B) no projection of host crops was computed.

Conclusions

Refer to main findings, avoid inclusion of control, is speculative.

Author Response

Dear Editors and Reviewers:

Thank you for your letter and for the reviewers' comments concerning our manuscript entitled "Predicting the current and future distributions of Frankliniella occidentalis (Pergande) based on the MaxEnt Species Distribution Model"(Insects-2320869). Those comments are all valuable and very helpful for revising and improving our paper, as well as the important guiding significance to our research. We have studied comments carefully and have made correction which we hope to meet with approval. The main corrections in the paper and the responds to the reviewer's comments are as flowing:

Reviewer

Major comments,

1). Major issues: "distribution is repetead.”(line 12)

Author's response: Thanks for your valuable comment. I have modified the expression to change distribution to suitable survival range.

2)"It does not indicate any clear result for future distribution, it enlarges, shorten or remain the same?”(line 16)

Author's response: Thanks for your meaningful comment. The results of different changes in the future have been added to the three distributions."In particular, the trend of its future geographical distribution is derived based on the results of future projections. Besides, it shows that the area of high and low suitability zones will continue to decrease in the future, and the area of medium suitability zones will significantly increase. ”(line 13 to 16)

3)"you mean variables?”(line 16)

Author's response: Thanks for your valuable advice. The word "environmental elements" in the original has been corrected to "environmental variables".

4). "You are referring to the virus, the disease is tomato spotted wilt.”(line 24)

Author's response: Thanks for your meaningful comment. It is my inaccurate expression.The sentence "The damage is primarily in the form of mechanical damage caused by adults and larvae and transmitted viral diseases" has now been corrected to "The damage is mainly divided into two aspects: mechanical damage caused by adults and larvae and the transmission of tomato spotted wilt virus (TSWV)".(line 24 to 25)

5) "wrong name. suborders are Terebrantia and Tubulifera (Mound & Morris, 2007)”(line 42)

Author's response: Thanks for your valuable comment. Now corrected to Terebrantia.(line 42)

6) "no need to keep the author component in the scientific name after first mention

Author's response: Thanks for your important suggestion. I'm very sorry for the oversight and have now corrected it in the full article.

7) "statement confuse, to what worm refers to? Not the development stage cause it is wrong. There are no worm stage in Thysanoptera.”

Author's response: Thanks for your meaningful suggestion. The sentence "The more serious damage to the worm than mechanical damage to plants is the spread of the tomato spotted wilt virus (TSWV)" has now been corrected to "However, the more serious damage caused by this insect is reflected in the transmission of tomato spotted wilt virus (TSWV) compared to the mechanical damage caused to plants".(line 58 to 60)

8)"repeated statement

Author's response: Thanks for your valuable comment. "The more serious damage to the worm than mechanical damage to plants is the spread of the tomato spotted wilt virus (TSWV).”The corresponding part of the original text has been deleted.

9) "what is the reference to this? You mean that TSWV affects severely the thrips populations? Explain.

Author's response: Thanks for your meaningful comment. My statement was not accurate. What I was trying to say was that not all F. occidentalis transmit TSWV, and that the few remaining infectious F. occidentalis can still cause widespread transmission of the virus. has been revised in the text to read "Despite the fact that only adults infected with TSWV in the larval stage are infectious, F. occidentalis is still highly transmissible to TSWV."(line 62 to 63)

10) "Too long paragraph to stress economic importance of this insect pest. Better include works related to species distribution of this pest.

Author's response: Thank you for your valuable advice. Excessive descriptions of TSWV in the original article have been removed."The characteristic symptom of TSWV is the appearance of distinct concentric rings on the leaves and fruits, which subsequently turn brown Premature decay can also occur, starting with older leaves, which fade to green and then take on a bronze color. Long-term infection can lead to the death of the entire plant, posing a serious threat to the planting and harvesting of the crop" has been removed. Since there are few previous studies on the distribution of F. occidentalis, no relevant information can be presented here.

11) "web sites does not match with references... Describe relationship.”(line 99)

Author's response: Thanks for your important suggestion. This paragraph is really not very relevant. After consideration, "Maxent is also used by government and non-governmental organizations for large-scale, real-world biodiversity mapping applications, including the Point Reyes Bird Observatory online application (http://www.prbo.org/) and the Atlas of Living Australia (http://www.ala.org.au/)" has been removed from the text.

12)"Regardless, you can mention the work of Zhang et al. 2019.  https://doi.org/10.1002/arch.21613 and He et al. 2019.  https://doi.org/10.1111/1744-7917.12721.”(line 102)

Author's response: Thanks for your valuable suggestion. The corresponding references have been added here.

13) "This an empirical, data-driven research.”(line 105)

Author's response: Thanks for your valuable comment. This description has been added here.(line 105)

14) "What are you referring to with Specigenetic data?”(line 110)

Author's response: We deeply appreciate your valuable comments. My expression was not accurate. It refers to the latitude and longitude points of the geographical distribution of the species F. occidentalis. It has been adjusted to "Species distribution data".(line 110).

15) "Mention which specific model was used to make future projections”(line 129)

Author's response: Thanks for your valuable advice. The three future scenarios of RCP2.6, RCP4.5 and RCP8.5 described here, its related data are only needed to obtain future climate projections by CMIP5 (IPCC Fifth Assessment Report, AR5). It has been presented in this paper.(line 129)

16) "Just list the metrics used, no need to explain how it works, instead, make reference to the original paper(s).”(line 148)

Author's response: Thanks for your meaningful suggestion. Because there are 19 Bioclimatic variables in total, only 5 variables were finally filtered out.The description of the selection process had to be made here for the sake of brevity in the later section. And relevant references have been added.(line 148)

17) "no need to mention if not used.”(line 170)

Author's response: Thanks for your valuable advice. The redundant part has been removed.

18)"AUC range is theoretically between 0 and 1”(line 174 to 176)

Author's response: Thanks for your valuable advice.The AUC standard has been corrected in the corresponding position.(line 174 to 176)

19) "According to whom?”(line 177)

Author's response: Thanks for your meaningful suggestion. The corresponding reference for the AUC standard has been added here.(line 177)

20)"This content is how maxent works but not related to what you obtained from the analysis. Just make reference to the original paper.”(line 183)

Author's response: Thanks for your meaningful suggestion.This section has been trimmed accordingly.(line 183)

  • "Mantain consistency in names of variables uppercase or lowercase but don't mix

Author's response: Thanks for your meaningful suggestion.Corrections have been made in the appropriate places throughout the text.

22) "do not duplicate info, delete table 2, expand and keep results in text.(line 170)

Author's response: Thanks for your valuable advice. Table 2 has been removed from the text.

23)"this is a ROC, AUC is the value; besides, it is an average of bootstrap sampling.”(Figure 1)

Author's response: Thanks for your valuable advice.The title in Figure 1 has been corrected to "Receiver Operating Characteristic Curve and AUC result of Maxent modelling."

  • "Where is bio4?(Figure 2)

Author's response: Thanks for your valuable advice. Bio 4 has now been added to Figure 2.

  • "from ... to ..., not at.”(line 227 to 233)

Author's response: Thanks for your valuable advice. Syntax changes have been made to the corresponding sections.(line 227 to 233)

  • "move to M&M.(line 148 to 154)

Author's response: Thanks for your valuable advice.Have moved "Based on the model-generated fitness index, the suitable distribution area of F. occiden-talis is divided into 4 levels, unsuitable area; moderately suitable area; poorly suitable area; highly suitable area. A 10-fold repeat cross-validation is then performed to run the Maxent software to prevent random errors from the predicted sample. Finally, the reclassification function in ArcGIS software is used to obtain the distribution level of F. occidentalis in the national suitable area, as shown in Figure (3). (line 148 to 154)

  • "round percent values to one decimal.”

Author's response: Thanks for your valuable advice.All percentage values in this paper have been rounded to one decimal place.

  • "confuse notation, as annotated it should be 40km^2?. Use scientific notation.(Table 4)

Author's response: Thanks for your valuable comment.All errors in this article have been corrected to "104km2".

  • "ibid(Table 4)

Author's response: Thanks for your valuable comment.But, I'm sorry I don't quite understand what "ibid" means.

  • "This is not relevant for the work. Extensive research using maxent + ArcGis has been performed previously.”

Author's response: Thanks for your valuable comment. The sentence "In this study, the Maxent model was integrated with ArcGIS software for the first time for the prediction of the potential distribution of F.occidentalis(Pergande) in China for both current and future periods" has been changed to "In this study, the Maxent model was first employed to predict the potential distribution of F. occidentalis in China for current and future periods". (line 283 to 284)

  • "This paragraph segment does not relate to the objectives of the paper.”(line 283)

Author's response: Thanks for your valuable comment.The corresponding redundant parts have been removed. (line 284)

  • Maxent has been tested with species having few records. This is not true.

Author's response: Thanks for your valuable comment.The wrong part of the discussion section has been removed.

  • "You mentioned the opposite in a previous paragraph.(line 289 to 290)

Author's response: Thanks for your valuable comment.It is my expression that is the problem. What I wanted to express is that MAXENT has the ability to make predictions with a few species occurrence points. However, the more species occurrence points that are available, the more reliable the prediction is, and the richer the species occurrence points of F. occidentalis are, the better it is for prediction.The phrase "leaving some limitations in the selection of predicted species" in (line 289 to 290) has been adjusted to "making further consideration needed in the selection of predicted species". It's just an aspect to consider, not a limitation of MAXENT.

  • "This is not how how predictions are checked, only suggest that similar conditions may remain in the future.”(line 311)

Author's response: Thanks for your valuable comment. Corrected "This coincides with the current pest-prone areas, indicating the high reliability of the predictions," changed to "This corresponds to the current area of frequent insect infestations" in the original. (line 311)

  • "Better discuss how projected areas in the future can be managed to minimize damage.”(line 336 to 346)

Author's response: Thanks for your valuable advice.Methods of biological control are enriched in the corresponding section, and literature is added. (line 336 to 346)

  • "you said no research on this topic have been performed...”(line 348)

Author's response: Thanks for your valuable advice.The incorrect statement has been removed.

  • "do not repeat results.”

Author's response: Thanks for your valuable advice.The repetition of "The AUC of this model is as high as 0.888, which indicates that the modeling performance is excellent and the results are highly reliable." has been deleted from the conclusion.

  • "Rephrase statement, findings just suggest that climatic conditions may not change so to shift future distribution of the pest.”(line 357 to 361)

Author's response: Thanks for your valuable advice.Supplemented with "In recent years, there have been frequent occurrences of high temperatures during summer in China, which is expected to have some impact on this insect. However, due to the biological characteristics of F. occidentalis and its R-strategy population, the impact of high temperature stress on the distribution of F. occidentalis in the next few years may not be significant. However, in the long term, temperature increases will lead to the expansion of F. occidentalis distribution area”. (line 357 to 361)

We tried our best to improve the manuscript and made some changes in the manuscript. These changes will not influence the content and framework of the paper.

We appreciate for Editors and Reviewers' warm work earnestly and hope that the correction will meet with approval.

Once again, thank you very much for your comments and suggestions.

Regards,

Zhiling Wang

College of Life Science, China West Normal University,

1 Shida Road, Nanchong, 637002, China

[Email] [email protected]

Reviewer 2 Report

Dear Authors,

The subject matter of the article is up-to-date and practically relevant. The data presented in the article are new. The introduction is well and thoroughly described. I only lack information on supplementing the bibliography regarding the bionomics of F. occidentalis. Repetition of the information from the first part into the abstract. I think the first paragraph is redundant.

To be completed: Western thrips up to a year 2006 was an A2 listed quarantine pest (OEPP/EPPO 1989).

There is no reference to research by other authors in the discussion. It seems that the indicated predictions should take into account regional factors not only abiotic but also biotic factors. There is little information about this in the text. This information could be used to supplement biological methods against thrips. Suggestion: it may be worth supplementing the Discussion chapter with information on biological methods of combating F. occidentalis.

In conclusions, it is worth referring to your own results and answering the question; How this trend is likely to develop in the coming years (short-term forecast based on time series extrapolation).

Yours sincerely,

Author Response

Dear Editors and Reviewers:

Thank you for your letter and for the reviewers' comments concerning our manuscript entitled "Predicting the current and future distributions of Frankliniella occidentalis (Pergande) based on the MaxEnt Species Distribution Model"(Insects-2320869). Those comments are all valuable and very helpful for revising and improving our paper, as well as the important guiding significance to our research. We have studied comments carefully and have made correction which we hope to meet with approval. The main corrections in the paper and the responds to the reviewer's comments are as flowing:

Reviewer

Major comments,

1)" Repetition of the information from the first part into the abstract. I think the first paragraph is redundant.

Author's response: Thanks for your valuable comment. Does the "the first part" above refer to the "Simple Summary" section? If so. This is the part where "INSECTS" is required to be written for a non-specialist audience, i.e. there are no technical terms without explanation, no references cited, and no abbreviated abstracts. So there is some duplication between "Simple Summary" and "abstract".

  • "To be completed: Western thrips up to a year 2006 was an A2 listed quarantine pest (OEPP/EPPO 1989).

Author's response: Thanks for your valuable comment.I'm very sorry, I don't quite understand about "Western thrips up to a year 2006 was an A2 listed quarantine pest (OEPP/EPPO 1989)". Is there any adjustment I need to make, or do I need to add it to some part of the article?

  • "There is no reference to research by other authors in the discussion. It seems that the indicated predictions should take into account regional factors not only abiotic but also biotic factors. There is little information about this in the text. This information could be used to supplement biological methods against thrips. Suggestion: it may be worth supplementing the Discussion chapter with information on biological methods of combating occidentalis. In conclusions, it is worth referring to your own results and answering the question; How this trend is likely to develop in the coming years (short-term forecast based on time series extrapolation).”

Author's response: Thanks for your valuable advice.Additions have been made to the discussion section on methods of biological control of F. occidentalis, and to the literature. (line 336 to 346) The possibilities for the development of F. occidentalis in the coming years are also added in the conclusion section."In recent years, there have been frequent occurrences of high temperatures during summer in China, which is expected to have some impact on this insect. However, due to the biological characteristics of F. occidentalis and its R-strategy population, the impact of high temperature stress on the distribution of F. occidentalis in the next few years may not be significant.”(line 357 to 361)

We tried our best to improve the manuscript and made some changes in the manuscript. These changes will not influence the content and framework of the paper.

We appreciate for Editors and Reviewers' warm work earnestly and hope that the correction will meet with approval.

Once again, thank you very much for your comments and suggestions.

Regards,

Zhiling Wang

College of Life Science, China West Normal University,

1 Shida Road, Nanchong, 637002, China

[Email] [email protected]

Reviewer 3 Report

Please see comments and suggestions on the attached file.

Author Response

Dear Editors and Reviewers:

Thank you for your letter and for the reviewers' comments concerning our manuscript entitled "Predicting the current and future distributions of Frankliniella occidentalis (Pergande) based on the MaxEnt Species Distribution Model"(Insects-2320869). Those comments are all valuable and very helpful for revising and improving our paper, as well as the important guiding significance to our research. We have studied comments carefully and have made correction which we hope to meet with approval. The main corrections in the paper and the responds to the reviewer's comments are as flowing:

Reviewer

Major comments,

  • "Both the summaries mention temperature and precipitation are vital directions, however, the direction in which these variables affect the distribution of Frankliniella occidentalis are not discussed so perhaps this could be rephrased, or the direction of the effect (i.e., increased temperature leads to greater spread) mentioned.”

Author's response: Thanks for your meaningful suggestion.The direction of the impact is added in the conclusion section."In recent years, there have been frequent occurrences of high temperatures during summer in China, which is expected to have some impact on this insect. However, due to the biological characteristics of F. occidentalis and its R-strategy population, the impact of high temperature stress on the distribution of F. occidentalis in the next few years may not be significant. However, in the long term, temperature increases will lead to the expansion of F. occidentalis distribution area.”(line 357 to 361)

  • "Lines 57 to 59 are a repeated sentence that refers to ‘worm’, but it is unclear what worm this refers too.”

Author's response: Thanks for your meaningful suggestion.The duplication has been removed. "The more serious damage to the worm than mechanical damage to plants is the spread of the tomato spotted wilt virus (TSWV). " Corrected to "However, the more serious damage caused by this insect is reflected in the transmission of tomato spotted wilt virus (TSWV) compared to the mechanical damage caused to plants."(line 58 to 60)

  • "Lines 88 to 95 compare the selected approach (MaxEnt) to other species distribution modelling approaches available, however no other approaches are specifically named and no examples of how MaxEnt consistently competes with other methods are cited. Given in recent years many studies have moved to machine learning based approaches it would be beneficial for this part of the introduction to discuss these approaches too and how MaxEnt compares to them.”

Author's response: Thanks for your meaningful suggestion.The main point here is the superiority of MAXENT compared to other models, and its superiority is that it is competitive. As suggested, the following is added to the text:"In the mid-80s of the 20th century, with the gradual development of SDM technology, there was a quantitative basis for the analysis of SDM in animals and plants for the first time. Among them, the more critical of these, the Maxent model, has shown sig-nificant superiority over other models in subsequent practice. The Maxent model is able to analyze the effects of different environmental variables on their survival dis-tribution for the studied species through a unique response curve. The Maxent model yields excellent statistics, benefiting from the spatial autocorrelation of the data, al-lowing its complex response curves to be fitted and the model to be remarkably close to the calibration data”. And the corresponding references as a basis.(line 74 to 82)

  • "No mention in ‘Specigenetic data’ (lines 109 to 122) of sample size and locations of occurrences. Also, how were latitude and longitude obtained using Google Earth based on locations provided? My concern is that this could give imprecise presence locations, especially as the bioclimatic variables could be as specific as 1km2.”

Author's response: Thanks for your meaningful suggestion. The term "Specigenetic data" has been corrected to "Species distribution data".The "sample size and locations of occurrences" are presented in the "Data Availability Statement".For "Google Earth", enter the geographic location mentioned in the literature and you will get a latitude and longitude coordinate.

  • "Were presence points manually cropped to ensure "the obtained occurrence points are necessarily smaller than its actual distribution area"(line 120 to 121), or is this an assumption based on limited data availability?”

Author's response: Thanks for your meaningful suggestion.In this study, ENMTools version 1.0.4 of the R platform was used to crop according to longitude."In this paper, we use ENMTools version 1.0.4 of the R platform to trim according to longitude, and because the small volume of F. occidentalis increases the difficulty of observation, the occurrence points obtained are necessarily smaller than its actual distribution area"has been added to the text where appropriate. (line 120 to 123)

  • "Include spatial resolution of bioclimatic variables used.”

Author's response: Thanks for your meaningful suggestion.The spatial resolution of variables was 2.5 arc-minute (approximately 4.5 km2).Added in the text where appropriate. (line 136)

  • "Include further information on the MaxEnt settings used for the model and why these settings were selected (e.g., number of background points and how they were generated).”

Author's response: Thanks for your meaningful suggestion.Some specific details about the settings when modeling MAXENT. In this paper, an analysis setup combining linear features, hinge features, and auto features is chosen. Max muber of background points is set to 10000, replicates is set to 10 times, and replicates run type is selected as bootstrap. Such a setting is beneficial for the optimization of the MAXENT model. At the same time, it proved to be reliable. It has been added to the corresponding position. "In this paper, the complexity of the model is controlled by adding parameters to the ENMTools function. The complexity of the Maxent model depends on two parameters: the model features and regularization of the parameters setting. The Maxent model is based on a set of "features" such as linear (L), hinge (H), quadratic (Q), threshold (T), and product (P). Given site-specific environmental values, the model is helped to determine the predicted probability of existence or the rate of environmental suitability of the study area. The feature combination (FC) settings for this study start with the default "LQ" training model, which is optimized to the "LQH" training model. The regularization multiplier (RM) value was optimized from the default value of 1 to 3.0, and the AUC value was optimized from 0.742 under the default setting to 0.888 now.”(line 202 to 212)

  • "Include justification of why only bioclimatic variables are used in modelling when other work on insects has shown other variables can be important (e.g., NDVI and landcover) when predicting distributions.”

Author's response: Thanks for your meaningful suggestion.In this paper, only climate is considered because he is a relatively important variable, and later work will consider soil, host and other directions. Instructions have been given in the text."The combination of abiotic and biotic factors will be considered in future prediction work, and this is an area that deserves attention and reflection in future forecasting efforts.”(line 294 to 296)

  • "AUC is vulnerable to inflation if testing datasets are not carefully selected. How does Maxent compute AUC and how does it mitigate this issue?”

Author's response: Thanks for your meaningful suggestion.In this paper, we set 10 replicates, which can effectively avoid this error."Figure (1) is the ROC curve of the same data used to estimate the model prediction accuracy, and the number of replicates of the first run of the model is 10 times.”(line 199 to 200) The MAXENT model was also optimized, and the optimized value was evaluated as "Excellent". "the AUC value was optimized from 0.742 under the default setting to 0.888 now.”(line 211 to 212)

  • "Table 1 is probably not needed in the text as the methods section already describes this information on lines 165 to 167.

Author's response: Thanks for your valuable advice.Table 1 has now been removed from the body.

  • "It is a little unclear how there can be both ‘no collinearity or low collinearity’ (line 195) and ‘significant correlation at .01 and .05 levels’ (table 2) and how this correlation will affect what can be inferred from variable importance.”

Author's response: Thanks for your valuable advice.Multicollinearity refers to the situation in multiple linear regression where the regression estimates are inaccurate due to the presence of a high correlation between the independent variables. If two variables are highly correlated, it indicates that the coefficients are very sensitive and that there is an error in the estimation of the coefficients. The emphasis is on the relationship between the variables and each other.In this paper, the conclusion of "no collinearity or low collinearity" verifies that the variables, after being filtered, do not affect each other's results. The "significant correlation at 0.01 and 0.05 levels" describes the correlation between a single variable and the distribution of the species, so the more significant the better. We need variables that have a significant effect on the distribution of the species but that do not interfere with each other at the same time.

  • "Table 2 shows that bio4 (temperature seasonality) is the second highest contributing variable, but this variable is excluded from later analysis when discussing the response of species to change (lines 210 to 227 / figure 2). The text would benefit from explaining why this variable has been excluded.”

Author's response: Thanks for your valuable advice.Because bio 4 is seasonal temperature variation. Although it is the second contributing variable to F. occidentalis, it is more reflective of changes in the distribution of F. occidentalis from season to season between one year. This paper focuses on the overall changes in the next three periods, so bio 4 is not explored in depth here.

  • "Future scenarios are described and shown well, though would also benefit from the MESS analysis mentioned above. Some further information on what the difference is between the three RCP scenarios would also be useful for the reader.”

Author's response: Thanks for your valuable advice.The corresponding content has been added. "RCP2.6 is a very mild scenario representing a future with low emissions and a focus on sustainable development. RCP4.5 is a moderate emission scenario representing a rela-tively mild future. RCP8.5 is a high emission scenario representing a future highly de-pendent on fossil fuels.”(line 132 to 135)

  • "If the current spread and distribution of occidentalis is primarily driven by synthetic pesticides (line 337), how well can the presence points used in this study reflect distribution based on environmental suitability? This needs further explanation in the discussion, or justification earlier in the text (e.g., methods) as if presence points are largely based on where pesticides have not been sprayed then large areas of suitable habitat may be excluded from the model.”

Author's response: Thanks for your valuable advice. The distribution points of F. occidentalis are randomly selected and whether or not pesticides are sprayed is also random. The details have been added in the text. "However, it should be noted that the selection of distribution points for the pest was randomly obtained from the Global Biodiversity Information Facility (http://www.gbif.org) and literature. The impact of pesticides on its distribution is also random, so distribution points are not primarily based on areas where pesticides have not been sprayed, and there is no exclusion of large suitable habitats that may have been left out of the model.”(line 324 to 329)

  • "More evidence is needed to say that the model is excellent and its results are highly reliable (line 357 to 358), such as how well the model can generalise to other areas and times, and separating the data into training and testing to show how well the model can predict the test data.”

Author's response: Thanks for your valuable advice.The AUC of this model is as high as 0.888, which indicates that the modeling performance is excellent and the results are highly reliable" in line 357 to 358 has been deleted because the first reviewer asked me not to repeat the results in the conclusion. But I am glad to provide you with evidence that the model is excellent. Our model has been optimized. The optimized RM, FC, Delta AICc, AUCDIFF, Mean AUC, Mean TSS, Mean Kappa were obtained. the results of each optimization can prove that the model is reliable. The specific results are shown in the following table:

Default

Optimization

RM

1

3.0

FC

LQ

LQH

Delta AICc

157.5

0

AUCDIFF

0.067

0.040

Mean AUC

0.742

0.888

Mean TSS

0.645

0.793

Mean Kappa

0.699

0.846

We tried our best to improve the manuscript and made some changes in the manuscript. These changes will not influence the content and framework of the paper.

We appreciate for Editors and Reviewers' warm work earnestly and hope that the correction will meet with approval.

Once again, thank you very much for your comments and suggestions.

Regards,

Zhiling Wang

College of Life Science, China West Normal University,

1 Shida Road, Nanchong, 637002, China

[Email] [email protected]

Round 2

Reviewer 1 Report

While current version addressed some comments, errors and suggestions of previous revision, some of them were not corrected and there is no document that explain why those suggestions were not taken into account. For example, current version still does not describe which model was used to estimate future projections, it only mention from the CMIP5, which is, by the way, an outdated project, current project is CMIP6 from which several specific models have been used to generate future bioclimatic layers.  

It still need a review by a native speaker, I cannot highlight every error in redaction.

It is usually recommended to use full scientific name at the begining of a sentence.

Thrips are hemimetabolous insects they do not hace a larval stage, they have egg, nymph and adult stages.

I suggest authors attend all the suggestions of current and previous version or else, include a rebuttal.

It is not clear how suggestion of using natural enemies to control the target pest is linked to findings of current and future distribution.

Additional comments are included in the manuscrit

Author Response

Dear Editors and Reviewers:

Thank you for your letter and for the reviewers' comments concerning our manuscript entitled "Predicting the current and future distributions of Frankliniella occidentalis (Pergande) based on the MaxEnt Species Distribution Model"(Insects-2320869). Those comments are all valuable and very helpful for revising and improving our paper, as well as the important guiding significance to our research. We have studied comments carefully and have made correction which we hope to meet with approval. The main corrections in the paper and the responds to the reviewer's comments are as flowing:

Reviewer

Major comments,

1) "For example, current version still does not describe which model was used to estimate future projections, it only mention from the CMIP5, which is, by the way, an outdated project, current project is CMIP6 from which several specific models have been used to generate future bioclimatic layers. "

Author's response: Thanks for your valuable comment. For future climate projections, we use the atmospheric circulation model CCCma_CanESM2. This has been added in the corresponding section of the text. " To estimate the impact of possible future climatic conditions on the distribution of F. occidentalis, the atmospheric circulation model CCCma_CanESM2 was used in the CMIP5 to simulate the RCPs. "(line 123 to 125)Regarding CMIP5, it still has some advantages. " Not all CMIP6 models of gcm showed significant improvements over CMIP5 models in future climate predictions. The ACC1 and MIR1 models in CMIP6 do not simulate the historical annual maximum and minimum temperatures as well as the CMIP5 model. Similarly, the GFD2, IPSL, and MIR1 models in CMIP6 simulate their historical solar radiation less well than the CMIP5 model. The simulation performance of the CMIP6 gcm model is lower than that of the CMIP5 model. The uncertainty is higher in the CMIP6 scenario compared to the CMIP5 RCP scenario. "(Li, Xinlin & Tan, Lili & Li, Yingxuan & Qi, Junyu & Feng, Puyu & Li, Baoguo & Zhang, Xueliang & Marek, Gray & Zhang, Yingqi & Liu, Haipeng & Srinivasan, Raghavan & Chen, Yong. (2022). Effects of global climate change on the hydrological cycle and crop growth under heavily irrigated management – A comparison between CMIP5 and CMIP6. Computers and Electronics in Agriculture. 202. 107408. 10.1016/j.compag.2022.107408.)However, in our future work we will heed the recommendation to use CMIP6 for future projections.

2)" It still need a review by a native speaker, I cannot highlight every error in redaction. "

Author's response: Thanks for your meaningful comment. The full text has now been checked for grammar and improved in detail.

3)" Rephrase statement, findings just suggest that climatic conditions may not change so to shift future distribution of the pest."

Author's response: Thanks for your valuable advice. The incorrect statement has been removed and rewritten. "It is expected to provide some reference for the prediction, forecast, and effective pre-vention and control of F. occidentalis. "(line 359 to 360)

4). "Full name if written at the begining of a sentence. "

Author's response: Thanks for your meaningful comment. Correction of the full text has been completed.

5) "Maxent DOES NOT compute survival, it estimates potential distribution, occurrence. "

Author's response: Thanks for your valuable comment. Sorry, I misrepresented it. It has been corrected to "suitable distribution".(line 73)

6) "This is wrong; using different concentration pathways just serve to estimate how different concentrations of GHG may affect bioclimatic variables. To measure model uncertainty other methods are available. "

Author's response: Thanks for your important suggestion. Sorry, this is my problem, and the wrong statement has been removed.(line 280)

7) "You took a random sample from the GBIF data set? Why? Redaction is confuse. "

Author's response: Thanks for your meaningful suggestion. Sorry for my inaccurate presentation. Our data are downloaded in their entirety from the GBIF with species occurrence records, but are filtered accordingly within each raster during the modeling process.(line 318 to 321)

8) " who say that pesticides are randomly distributed? Pesticides are applied where needed, farmers do not apply pesticides just because. "

Author's response: Thanks for your valuable comment. After consideration, you are correct. Therefore, the expressions have been removed from the original text.(line 323)

9) "How do you relate the distribution of the pest with the natural enemies action? "

Author's response: Thanks for your meaningful comment. The relevant expressions have been added to the text. "Specifically, the future range of natural enemies can be predicted and then compared to the future range of F. occidentalis. Biological control can be carried out by expanding the suitable distribution area of natural enemies where both are present. "(line 335 to 338)

10) "You didnt mention or discuss what competitive strategy this pest may adopt. "

Author's response: Thank you for your valuable advice. F. occidentalis's competitive strategy has now been added to the introduction. "as well as the R-strategy adopted by the population"(line 53)

11) "Thrips are hemimetabolous insects they do not hace a larval stage, they have egg, nymph and adult stages. "

Author's response: Thanks for your important suggestion. The misrepresentation of "larval" has now been removed from the entire text.

We tried our best to improve the manuscript and made some changes in the manuscript. These changes will not influence the content and framework of the paper.

We appreciate for Editors and Reviewers' warm work earnestly and hope that the correction will meet with approval.

Once again, thank you very much for your comments and suggestions.

Regards,

Zhiling Wang

College of Life Science, China West Normal University,

1 Shida Road, Nanchong, 637002, China

[Email] [email protected]